

# Macrophage polarization in tissue fibrosis

Huidan Yang, Hao Cheng, Rongrong Dai, Lili Shang, Xiaoying Zhang and Hongyan Wen

Department of Rheumatology, Shanxi Medical University Second Affiliated Hospital, Taiyuan, Shanxi Province, China

## ABSTRACT

Fibrosis can occur in all major organs with relentless progress, ultimately leading to organ failure and potentially death. Unfortunately, current clinical treatments cannot prevent or reverse tissue fibrosis. Thus, new and effective antifibrotic therapeutics are urgently needed. In recent years, a growing body of research shows that macrophages are involved in fibrosis. Macrophages are highly heterogeneous, polarizing into different phenotypes. Some studies have found that regulating macrophage polarization can inhibit the development of inflammation and cancer. However, the exact mechanism of macrophage polarization in different tissue fibrosis has not been fully elucidated. This review will discuss the major signaling pathways relevant to macrophage-driven fibrosis and profibrotic macrophage polarization, the role of macrophage polarization in fibrosis of lung, kidney, liver, skin, and heart, potential therapeutics targets, and investigational drugs currently in development, and hopefully, provide a useful review for the future treatment of fibrosis.

## INTRODUCTION

Macrophages are a class of pluripotent and highly plastic immune cells that play a key role in host defense (*Hirayama, Iida & Nakase, 2017*), tissue repair and regeneration (*Yu et al., 2022*), as well as fibrosis (*Shapouri-Moghaddam et al., 2018*; *Wynn & Vannella, 2016*). Macrophages can be bone marrow derived (*i.e.*, Bone marrow derived macrophages or BMDM), or tissue resident (*i.e.*, tissue resident macrophages or TRM) (*Davies et al., 2013*; *Ginhoux & Jung, 2014*; *Locati, Curtale & Mantovani, 2020*). In different tissue environments, they can be polarized into two primary macrophage subpopulations: (1) classically activated or M1 macrophages that are activated by LPS or Th1-type cytokines such as IFN-$\gamma$. M1 macrophages release pro-inflammatory cytokines like TNF-$\alpha$, IL-6, and IL-1$\beta$ to induce an inflammatory response (*Patel et al., 2017*). Moreover, M1 macrophages highly express CD80/CD86 and nitric oxide synthase (iNOS); (2) the alternatively activated or M2 macrophages that can be activated by Th2-type cytokines such as IL-4 and IL-13, and highly express CD206/CD163. M2 macrophages promote tissue repair, regeneration, and fibrosis by secreting multiple cytokines, in which TGF-$\beta$ and platelet-derived growth factor (PDGF) promote fibroblasts activation and myofibroblasts formation (*Wynn & Vannella,*

Corresponding author
Hongyan Wen,
wenhongyan0509@aliyun.com

*2016*). M2 macrophages can be further polarized into M2a, M2b, and M2c macrophages. M2a and M2c macrophages secrete TGF-$\beta$ and other pro-fibrotic factors to induce tissue fibrosis (*Perdiguero & Geissmann, 2016*). M2b macrophages, also known as regulatory macrophages, maintain a balance between pro-inflammatory and anti-inflammatory functions. Moreover, single-cell RNA sequencing(scRNA-seq) has identified several highly specific pro-fibrogenic macrophage subpopulations such as TREM2CD206, TREM2CD9 and highly expressing PLA2G7 macrophage (*Ramachandran et al., 2019*; *Satoh et al., 2017*; *Wang et al., 2022b*; *Wendisch et al., 2021*).

Fibrosis is a pathological process in that tissue repair becomes dysregulated following many types of tissue injury (*Pakshir & Hinz, 2018*). It is characterized by abnormal increase and excessive deposition of extracellular matrix (ECM) in tissues (*Henderson, Rieder & Wynn, 2020*). Fibrosis can occur in multiple organs such as the lung, kidney, liver, heart, skin, and other organs. The mild fibrosis has few abnormalities in the clinical examination, no clinical symptoms, and no significant impact on daily life; while the severe fibrosis shows tissue structure damage and organ sclerosis, eventually leading to organ failure. At present, there are few effective treatments, which place a heavy burden on humans, and about 45–50% of deaths can be attributed to fibrosis in developed countries alone (*Friedman et al., 2013*). Myofibroblasts undergo excessive proliferation and activation to produce ECM and collagen, which play an important role in the process of fibrosis. It is highly heterogeneous, because mesenchymal progenitor cells/stem cells (MSC), adipocyte progenitor cells (AP), epithelial cells, endothelial cells, pericyte, podocytes, and monocyte macrophages can transform into myofibroblasts (*Bhattacharyya, Wei & Varga, 2011*; *Campanholle et al., 2013*; *Micallef et al., 2012*). Macrophage-myofibroblast transition (MMT), a term coined by *Nikolic-Paterson, Wang & Lan (2014)*, is the transition of infiltrating bone marrow-derived monocytes into myofibroblasts in the injured kidney, as the cells in the process of transition can express markers of both lineages. Subsequent studies have shown that MMT is involved in progressive fibrotic disease and MMT cells were mainly of M2 phenotype (*Meng et al., 2016*; *Yang et al., 2021*). However, the underlying mechanisms of MMT are not well defined. It has been suggested that TGF-$\beta$/Smad and mineralocorticoid receptor (MR) activation may stimulate MMT (*Wang et al., 2016*). Epithelial-to-mesenchymal transition (EMT) is a process in which epithelial cells lose their epithelial features and acquire mesenchymal characteristics (*Stone et al., 2016*). Sustained EMT is a key mechanism underlying the fibrotic pathology of multiple organs including the skin (*Chapman, 2011*; *Kaimori et al., 2007*; *Postlethwaite, Shigemitsu & Kanangat, 2004*; *Zeisberg & Kalluri, 2008*). Endothelial-mesenchymal transition (EndoMT) is a complex biological process in which endothelial cells lose their endothelial markers, adhesion, and apical-basal polarity and transit into mesenchymal cell type under certain conditions, leading to organ fibrosis and dysfunction (*Dejana, Hirschi & Simons, 2017*; *Zeisberg et al., 2008*). TGF-$\beta$ and connective tissue growth fact (CTGF), which activate the Wnt/$\beta$-Catenin pathway, can promote pericyte and podocyte transition to myofibroblasts (*Dai et al., 2016*; *Lin et al., 2008*; *Ren et al., 2013*). Most importantly, macrophages provide the microenvironment for the proliferation and activation of myofibroblasts.

Previous studies have mostly focused on the mechanism of macrophages involved in inflammation and tissue damage repair. In recent years, more studies have shown that macrophage polarization plays a role in fibrosis of the lung, kidney, liver, skin, heart, and other organs (*Vannella & Wynn, 2017*; *Wynn & Barron, 2010*). After tissue damage, the M1 phenotype releases proinflammatory cytokines to involve in the initiation phase of inflammation. In contrast, the M2 phenotype is involved in the repair phase. A shift from M1 to M2 was described in the wound healing processes from inflammation to restoration (*Duffield et al., 2005*; *Gibbons et al., 2011*; *Lucas et al., 2010*). If the initial insult leads to sustained inflammatory or tissue repair imbalance, both can promote fibrosis. However, macrophages also produce matrix metalloproteinases (MMP) involved in the regression of fibrosis. Therefore, exploring the exact mechanism of macrophage phenotype transformation in fibrosis is helpful to provide a new therapeutic strategy.

The purpose of this review is (1) to summarize the major signaling pathways relevant to macrophage-driven fibrosis and profibrotic macrophages polarization; (2) to describe the role of macrophages in fibrosis of lung, kidney, liver, skin, and heart; (3) to discuss potential therapeutics targets and investigational drugs currently in development. It may be helpful to study macrophage function, fibrosis pathogenesis, and anti-fibrosis drug.

## MAJOR FIBROSIS-RELATED SIGNALING PATHWAYS

### TGF-$\beta$/Smad pathway

A TGF-$\beta$/Smad signaling pathway is one of the main pathways involved in fibrosis. TGF-$\beta$ superfamily mainly includes TGF-$\beta$1, TGF-$\beta$2, and TGF-$\beta$3, which are produced by macrophages, fibroblasts, alveolar epithelial cells, activated T cells, or B cells. Macrophage-derived TGF-$\beta$1 is typically profibrotic, and studies have identified various macrophage subsets as key producers of TGF-$\beta$1 (*Wynn & Barron, 2010*; *Zhu et al., 2017*). TGF-$\beta$ acts on type II receptors first and then binds to type I receptors to form a receptor complex, which leads to the phosphorylation of Smad2 and Smad3. It can activate transcription factors, and promotes collagen synthesis, ECM deposition, and cell transdifferentiation involved in tissue fibrosis (Fig. 1). Smad7 negatively regulates the TGF-$\beta$/Smad signaling pathway (*Hayashi et al., 1997*; *Heldin, Miyazono & Ten Dijke, 1997*). TGF-$\beta$/Smad signaling pathway can promote MMT and EMT (*Wang et al., 2016*; *Zhu et al., 2017*). TGF-$\beta$/Smad signaling pathway may be a potential anti-fibrosis therapeutic target. Fisetin (3,3′,4′,7-tetrahydroxyflavone), a dietary flavonoid, can alleviate renal fibrosis by inhibiting the phosphorylation of Smad3, and accumulation of profibrotic M2 macrophages (*Ju, Kim & Han, 2023*). LY2109761, a small molecule-selective TGF-$\beta$ receptor type I/II kinase inhibitor, can block M2 macrophages induce EMT by suppressing TGF-$\beta$-induced Smad2 phosphorylation signaling pathway in in-vitro experiments (*Kim et al., 2022*). Decreasing the number of TGF-$\beta$1-producing macrophages, rather than comprehensively attenuating TGF-$\beta$1 may provide a more rational approach to ameliorate fibrosis. Mannosylated albumin nanoparticles loaded with TGF-$\beta$1-siRNA specifically bind to the mannosylated receptor CD206 on the surface of M2 macrophages which silences the expression of TGF-$\beta$1 and significantly alleviate bleomycin-induced pulmonary fibrosis in mice (*Singh*

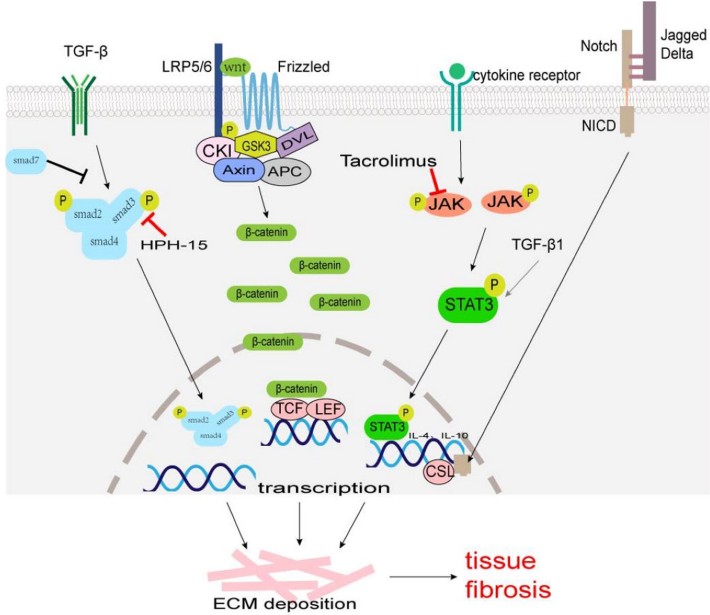

**Figure 1** **The major fibrosis-related signaling patnways.** Some signaling pathways are associated with tissue fibrosis, such as TGF-$\beta$/Smad, Wnt/$\beta$-Catenin, JAK/STAT3, and Notch which regulate target gene transcription, ECM production and transformation of different cells into myofibroblasts to lead to tissue fibrosis.

*et al., 2022*). Baccatin III (BAC), the precursor of the semisynthesis of paclitaxel, suppresses TGF-$\beta$1 production by macrophages (Table 1) (*Acharya et al., 2012*; *Nie et al., 2019*).

## Wnt/$\beta$-Catenin pathway

Wnt, a cell signaling molecule, can stimulate cell proliferation, differentiation, and migration. The Wnt/$\beta$-Catenin pathway is the classic Wnt signaling pathway. Wnt binds to its receptor Frizzled (an atypical G protein with seven trans-membrane domains) and co-receptor low-density lipoprotein receptor-associated protein 6 (LRP6) or LRP5, to activate Dishevelled (Dvl) leading to the phosphorylation of LRP5/6 and inhibiting the activity of $\beta$-catenin-degrading complexes formed by serine/threonine protein kinase (GSK3) and other proteins, stabilizing free $\beta$-Catenin in the cytoplasm (*MacDonald, Tamai & He, 2009*). $\beta$-Catenin accumulated in the cytoplasm enters the nucleus and binds to T cell factor (TCF)/lymphatic enhancer binding factor (LEF) to activate the transcription of target genes, such as c-Myc, Axin2, MMP7, Tcf, fibronectin and so on (Fig. 1) (*MacDonald, Tamai & He, 2009*). The expression products of these target genes (Tcf21, Sox2, and Snai2) can induce EMT to promote cardiac fibrosis (*Acharya et al., 2012*; *Tao et al., 2016*). The Wnt/$\beta$-catenin signaling pathway also regulates the differentiation of alveolar macrophages and promotes the occurrence of pulmonary fibrosis (*Sennello et al., 2017*).

## JAK/STAT3 pathway

Janus kinase (JAK)/signal transducer and activator of transcription (STAT) signal pathways were first identified in mammals, which regulate cell growth, proliferation, differentiation,

**Table 1  The drugs and mechanisms in blocking macrophage-driven fibrosis.**

| Drugs | Research objects | Mechanisms | Organs | Reference |
|---|---|---|---|---|
| RP-832c | Mice | M2 ↓; TGF-$\beta$1 | Lung | *Ghebremedhin et al. (2023)* |
| Microcystin-LR | Rats, cells | M2 ↓; TGF-$\beta$1 / Smad, EMT, FMT | Lung | *Wang et al. (2020)* |
| Tacrolimus | Mice | M2 ↓; JAK2/STAT3 | Lung | *Liu et al. (2022)* |
| Baccatin III(BAC) | Mice, cells | TGF-$\beta$1 ↓ | Lung | *Nie et al. (2019)* |
| Sart1 siRNA-loaded liposomes | Mice, cells | M2 ↓; STAT6 / PPAR-$\gamma$ | Lung | *Pan et al. (2021)* |
| IL-24 knockout | The serum of IPF, mice | M2 ↓; SOCS, STAT6 / PPAR-$\gamma$ | Lung | *Rao et al. (2021)* |
| Nanoparticles loaded with TGF-$\beta$1-siRNA | Mice, cells | TGF-$\beta$1 ↓ | Lung | *Singh et al. (2022)* |
| Ruxolitinib | Mice, cells | JAK | Lung | *Bellamri et al. (2023)* |
| Capsaicin | Mice | M1 ↓; Notch | Liver | *Sheng et al. (2020)* |
| Annexin A1 | Mice | Trem2CD9 macrophage to Kupffer cells | Liver | *Gadipudi et al. (2022)* |
| Emodin | Rats, cells | M1 ↓,M2 ↓; TGF-$\beta$, Notch | Skin | *Xia et al. (2021)* |
| CHRFAM7A | Mice | M2 ↓; Notch | Skin | *Li et al. (2020)* |
| Iguratimod | Mice, cells | TGF-$\beta$1 / Smad | Skin | *Xie et al. (2022)* |
| fisetin | Mice | M2 ↓; Smad3 | Kidney | *Ju, Kim & Han (2023)* |
| Cucurbitacin-B | Mice | M2 ↓; JAK2/STAT3 ↓ | Liver | *Sallam, Esmat & Abdel-Naim (2018)* |

and apoptosis (*Hou et al., 2002*). Many studies have confirmed that JAK/STAT3 signaling plays an important role in the pathogenesis of fibrosis, and the up-regulation of STAT3 expression has been detected in fibrotic tissues (*Ogata et al., 2006*; *Pechkovsky et al., 2012*; *Pedroza et al., 2016*). The phosphorylation of STAT3 can regulate the transcription of IL-4 and IL-10, and promote the polarization of M2 macrophages (Fig. 1) (*Sun et al., 2021*). STAT3 promotes the fibrosis process through the following ways: (1) inducing the production of ECM; (2) regulating the transcription of MMP and tissue inhibitors of metalloproteinases (TIMPs); (3) inhibiting the apoptosis of fibroblasts; (4) participating in the EMT process as a non-standard TGF-$\beta$1 downstream factor; (5) promoting M2 macrophage polarization. Some JAK/STAT3 inhibitors, such as dual inhibitor JSI-124; STAT3 inhibitors C188-9, S3I-201, and Cucurbitacin-B, have been shown to reduce fibrosis progression in preclinical bleomycin-induced mice models (*Chakraborty et al., 2017*; *Chakraborty et al., 2021*; *Milara et al., 2018*; *Pedroza et al., 2018*; *Sallam, Esmat & Abdel-Naim, 2018*; *Zhang et al., 2022*). Tacrolimus inhibits JAK2/STAT3 signaling by targeting the JAK2 protein in macrophages, thereby suppressing M2 macrophage polarization (Fig. 1). When the secretion of pro-fibrotic cytokines CD206, CD163, TGF-$\beta$, IL-10, and IL-12 are blocked, the conversion of fibroblast to myofibroblast is reduced, thus alleviating the progression of BLM-induced pulmonary fibrosis (*Liu et al., 2022*).

## Notch signaling pathway

Notch signaling occurs *via* cell–cell contact and is evolutionarily highly conserved (*Artavanis-Tsakonas, Rand & Lake, 1999*). The Notch family is made of four Notch
receptors (Notch 1–4) and five ligands (Jagged1 and 2, Delta-like1,3 and 4) in mammals (*Siebel & Lendahl, 2017*). Notch activation requires the binding between a Notch receptor and a Notch ligand on two different, neighboring cells. Notch intracellular domain (NICD) liberates from the plasma membrane by enzyme digestion and translocates to the cell nucleus. In the cell nucleus, NICD interacts with transcription factor CSL (such as Rbpj, CBF1, and so on) to regulate the expression of downstream genes (such as HEY, HES family, MYC, and so on) (Fig. 1) (*Siebel & Lendahl, 2017*). There was research showed that in human hepatic fibrosis biopsies, stronger Notch activation is correlated with more severe fibrosis (*He et al., 2015*). The Notch can exert a pro-fibrotic role in the lung, kidney, liver, and skin by regulating myofibroblast activation and EMT, or dialoguing with other potent fibrogenic pathways, in particular the TGF-$\beta$1 signaling (*Condorelli et al., 2021*; *Hu & Phan, 2016*). Notch signaling also plays an important role in the regulation of macrophage polarization and functions (*Chen et al., 2021*; *Li et al., 2023*; *Xu, Chi & Tsukamoto, 2015*). In a murine model of liver fibrosis infected with *Schistosoma japonicum* (a parasite that is endemic in Asia), inhibition of the Notch1/Jagged1 signaling pathway could reverse macrophage M2 polarization, thereby alleviating liver fibrosis (*Zheng et al., 2016*).

# MACROPHAGES POLARIZATION AND TISSUE FIBROSIS

## Macrophages and lung fibrosis

Idiopathic pulmonary fibrosis (IPF) is the prototypic progressive fibrosing interstitial lung disease (ILD). The median survival time of IPF is about 3–5 years after diagnosis, and patients with exacerbation of IPF have in-hospital mortality greater than 50% (*Spagnolo et al., 2021*). The pathogenesis of IPF remains obscure. At present, the FDA approved two drugs, nintedanib, and pirfenidone to treat pulmonary fibrosis. They can stabilize patients' conditions well but do not reverse the progression of fibrosis (*Glass et al., 2022*).

Pulmonary macrophages derive from monocytes and are widely present in the alveoli and lung interstitium. When epithelial cells are damaged, monocyte precursors are largely activated in the action of monocyte chemoattractant protein-1 (monocyte chemotactic protein 1, MCP-1) and enter the lungs to differentiate into alveolar macrophages aggregated to the site of inflammation (*Jiang et al., 1992*). Under the action of inflammatory factors such as LPS, and IFN-$\gamma$, they are polarized into M1 macrophages. They secrete TNF-$\alpha$, and IL-6, and play the role of promoting inflammation (*Routray & Ali, 2016*). *Sakaguchi et al. (2016)* also confirmed in the rat acute lung injury model induced by LPS, M1 macrophages secrete IL-23 to promote the proliferation of lung memory Th17 cells and induce the production of IL-17, IL-22, and IFN-$\gamma$, thus accelerating the process of lung injury. Moreover, M1 alveolar macrophages can also produce MMP to promote ECM degradation and participate in the regression of fibrosis (*Wynn, Chawla & Pollard, 2013*). The sustained inflammatory response will promote the occurrence of tissue fibrosis. Notably, abundant infiltration of M2 macrophages was detected in the lung tissues of IPF patients and bleomycin-induced pulmonary fibrosis model mice (*Wang et al., 2019*). M2 macrophages promote the generation of pulmonary fibrosis effector cells in the following ways: (1) Secreting TGF-$\beta$, IL-4, and IL-13 to transdifferentiate circulating fibroblasts into
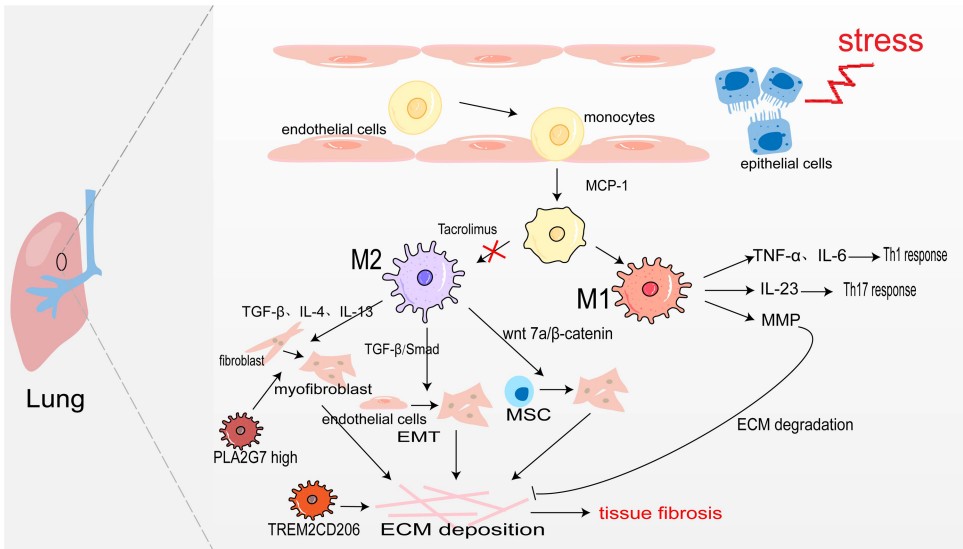

**Figure 2** **Macrophages in lung fibrosis.** After damage of alveolar epithelial cells, monocytes are activated by MCP-1 chemotaxis and further differentiated into M1 and M2 macrophages under the action of inflammatory factors. M1 macrophages secrete TNF-$\alpha$, IL-6 and IL-23 to promote inflammation, and also produce MMP to degrade ECM. The secretion of TGF-$\beta$, IL-4, IL-13 and Wnt7 by M2 macrophages makes fibroblasts, endothelial cells and MSC transdifferentiate into myofibroblasts, producing ECM and leading to pulmonary fibrosis. Profibrotic TREM2CD206 and PLA2G7[high] macrophage.

$\alpha$-SAM+ myofibroblasts (*Yao et al., 2015*); (2) TGF-$\beta$ activates Smad2/3 to promote EMT (*Tanjore et al., 2009*; *Zhu et al., 2017*); (3) Secreting Wnt7a protein to activate the Wnt/$\beta$-catenin channel and promote the differentiation of lung MSC to myofibroblasts (*Hou et al., 2018*). However, M2 macrophages may also directly promote pulmonary fibrosis through transdifferentiating into myofibroblast. In fibrotic lung tissue with unilateral ureteral obstruction (UUO) rats, *Yang et al. (2021)* used immunofluorescence staining as their bases for quantification and showed that approximately 30% of myofibroblasts were CD68+$\alpha$-SMA+ MMT cells, and up to 35% were co-expressing for M2 macrophage marker CD206(CD206+$\alpha$-SMA+). There was also a study showing that aldosterone may affect MMT by activating MR on the surface of macrophage (*Marzolla et al., 2014*). All in all, M2 macrophages promote myofibroblast formation to promote pulmonary fibrosis (Fig. 2).

In recent years, people taking advantage of scRNA-seq identified some disorder-specific profibrotic macrophage subtypes. In severe acute respiratory syndrome coronavirus 2 (SARS-CoV-2)-induced pulmonary fibrosis, the number of profibrotic TREM2CD206 macrophages is increased (*Wendisch et al., 2021*). Subsequently, a study showed that TREM2 silencing might alleviate pulmonary fibrosis possibly through inhibiting the secretion of profibrotic factors such as TGF-$\beta$ and PDGF and reducing macrophage polarization *via* regulation STAT6 activation (*Luo et al., 2023*). *Wang et al. (2022b)* identify a macrophage subpopulation highly expressing PLA2G7 in the fibrotic lungs, which promotes fibroblast-to-myofibroblast transition.
## Macrophages and kidney fibrosis

Kidney fibrosis is the ultimate common pathway of most progressive chronic kidney disease (CKD) (*Lv et al., 2018*). In 2017, 697.5 million cases of CKD were recorded, for a global prevalence of 9.1%; 1.2 million people died from CKD, and the global mortality rate increased by 41.5% compared to 1990 (*Collaboration GBDCKD, 2020*). Its pathological features are myofibroblast proliferation and activation, epithelial cell dysfunction, recruitment of circulating fibrocytes, excessive production, and deposition of ECM (*Liu, 2011*). At the early stage of kidney injury, a large number of chemokines represented by CCL2 are released locally to recruit CCR2+/Ly6C$^{high}$ monocyte/macrophages to the site of injury, and produce many inflammatory factors to trigger an inflammatory response (*Braga et al., 2018*). Studies have also shown that the accumulation of B cells in the early stage of kidney injury enhances the mobilization and recruitment of monocyte/macrophage cells, thus accelerating renal fibrosis (*Han et al., 2017*). Depletion of monocyte/macrophages through clodronate liposomes can lower blood pressure and reduce hypertensive kidney injury and fibrosis (*Huang et al., 2018*). *Hu et al. (2023)* also demonstrated that in the clodronate liposomes treatment group, IL-10, and TGF-$\beta$ expression was decreased, and TNF-$\alpha$ was not changed, which may attenuate renal fibrosis because of M1/M2 polarization. In the late repair stage of kidney injury, macrophages transform into M2 type participating in kidney fibrosis through the following ways: firstly, they release TGF-$\beta$, IL-1$\beta$, PDGF, and other pro-fibrosis factors to activate fibroblasts, produce ECM, and promote the occurrence of renal interstitium fibrosis. Secondly, M2 macrophages are directly involved in the process of kidney fibrosis by transforming into myofibroblasts through TGF-$\beta$/Smad3-mediated MMT. In the mouse model UUO and kidney biopsy samples from patients with chronic active renal allograft rejection, CD68+/$\alpha$-SMA + cells accounted for about 50% of the total number of $\alpha$-SMA + myofibroblasts, of which 75% were M2 type co-expressing CD206, and a small number were M1 type co-expressing iNOS (*Wang et al., 2017b*). On the contrary, a study has shown that only a small part of monocyte/macrophages transformed into myofibroblasts (*Kramann et al., 2018*). It deserves further study. Thirdly, macrophage-derived cytokines activate EMT and transdifferentiate pericytes into myofibroblasts (*Falke et al., 2015*). Fourthly, podocytes can obtain a mesenchymal property in a high-glucose condition and transdifferentiate into myofibroblasts (*Yamaguchi et al., 2009*). In elevated glucose levels, podocytes and pericytes also can secrete TGF-$\beta$1 to induce mesenchymal transition (*Wu et al., 2017*; *Xie et al., 2015*). Furthermore, there is evidence that pro-fibrotic macrophages participate in the regression of fibrosis by producing MMP to degrade ECM (Fig. 3) (*Anders & Ryu, 2011*). Ang II significantly increased monocyte/macrophage recruitment in the kidney and the expression of TGF$\beta$1, which is involved in renal fibrosis by promoting the differentiation of fibroblasts into myofibroblasts and ECM production (*Huang et al., 2018*; *Ruiz-Ortega et al., 2006*).

## Macrophages and liver fibrosis

Liver fibrosis is a pathological repair process for the formation of pseudolobules after hepatocyte destruction caused by chronic liver disease. Cirrhosis affects approximately

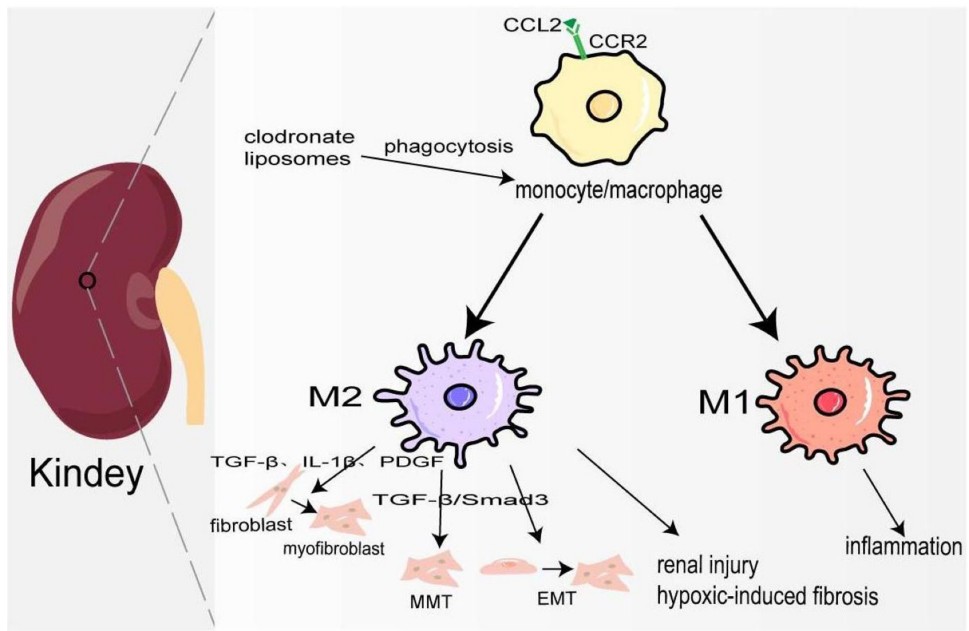

**Figure 3 Macrophages in kindey fibrosis.** After kidney injury, locally released CCL2 recruits CCR2+/Ly6C[high] monocyte/macrophages to the injured site, leading to local inflammatory response. Under the mediation of different signaling pathways macrophages are polarized to M1 and M2, which produce a variety of cytokines to maintain inflammation, activate fibroblasts, MMT, EMT and hypoxia-induced fibrosis.

2.2 million adults in the US (*Tapper & Parikh, 2023*). From 2010 to 2021, the annual age-adjusted mortality of cirrhosis increased from 14.9 per 100,000 to 21.9 per 100 000 people (*Tapper & Parikh, 2023*).

Hepatic macrophages, including tissue macrophages, namely hepatic Kupffer cells and BMDM, play their respective roles in different stages of hepatic fibrosis. Kupffer cells exist in the hepatic sinusoids which recognize, phagocytic, and eliminate foreign antigens, secrete inflammatory cytokines and chemokines to stimulate the body's inflammatory response, and recruit monocytes/macrophages. *Han et al. (2019)* found that compared with the normal control group, the expression of CD68 in fibrotic fatty hepatitis tissue was significantly increased, and all were GFP+, F4/80+, and Ly6C+ macrophages. The consumption of macrophages with chlorphosphonate liposomes could alleviate liver fibrosis, indicating that the macrophages involved in liver fibrosis are BMDM rather than Kupffer cells. Interestingly, the number of hepatic M2 macrophages is positively correlated with the severity of liver fibrosis, and the up-regulated expression of CCL promotes macrophages' conversion to the M2 type (*Xi et al., 2021*). On the one hand, BMDM produces IL-1$\beta$ and TNF-$\alpha$ to promote NF-$\kappa$B-mediated myofibroblast proliferation (*Pradere et al., 2013*). On the other hand, profibrotic TGF-$\beta$ activates the resting hepatic stellate cells (HSC) in a smad2/Smad3-dependent manner, and transforms HSCs into myofibroblasts, producing excessive ECM components and promoting the occurrence of liver fibrosis (*Xu et al., 2016*). It has recently been found that increased expression of

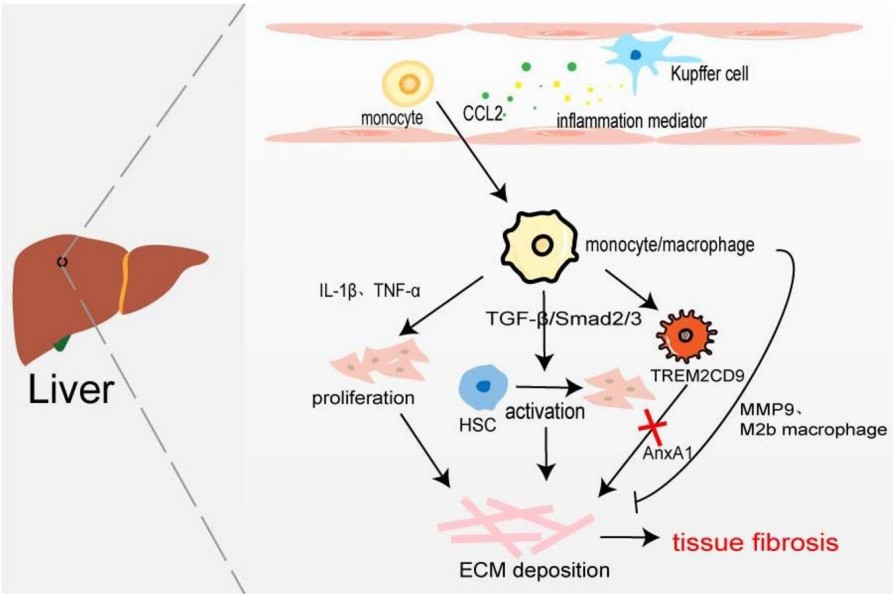

**Figure 4 Macrophages in liver fibrosis.** Kupffer cell is an inherent macrophage in hepatic sinusoids. When hepatocytes are damaged, inflammatory mediators such as chemokines are produced, and monocytes/macrophages are recruited into the tissues. The proliferation of myofibroblasts and the activation of resting HSC resulted in excessive deposition of ECM leading to liver fibrosis. A novel TERM2CD9 macrophage subsets promotes collagen formation by HSC and causes liver fibrosis. Moreover, monocytes/macrophages are also involved in fibrosis regression by secreting MMP9 or polarizing into M2b macrophages.

Mer tyrosine kinase regulates downstream STAT3, ERK1/2, and p38 phosphorylation, and promotes HSC migration and proliferation (*Pastore et al., 2022*). HSCs can secret a large amount of lactate to increase the levels of acetylation modification at the promoter regions of genes (Arg-1, CD163, IL-10, and TGF-$\beta$1), thereby promoting the transformation of macrophages from M1 type to M2 type and the progression of liver fibrosis (*Chen et al., 2022*). Furthermore, macrophages are involved in the regression of fibrosis by secreting MMP9 to degrade ECM or polarizing into M2b-like macrophages (Fig. 4) (*Wang et al., 2017a*). *Ramachandran et al. (2019)* taking advantage of scRNA-seq identified a novel scar-associated TERM2CD9 macrophage subset (SAMs), which expands in the human fibrotic liver. SAM subsets promote collagen formation by HSC and cause liver fibrosis (*Esparza-Baquer et al., 2021*). Infiltrating monocytes in liver fibrosis differentiate into SAMs in response to IL-17A, GM-CSF, and TGF-$\beta$ (*Fabre et al., 2023*).

Annexin A1 (AnxA1), a calcium-phospholipid-binding protein, can prevent the development of fibrosis in NASH by regulating liver macrophage differentiation from Trem2CD9 profibrotic macrophage to Kupffer cells (*Gadipudi et al., 2022*). Cenicriviroc (CVC), a CCR2 and CCR5 receptor antagonist, inhibits infiltrating monocyte-derived macrophages. In the CENTAUR phase IIb study, CVC 150 mg once daily (QD) improved fibrosis at month 12 and was twice as likely to provide antifibrotic benefit *vs* placebo (20% *vs* 10%) (*Friedman et al., 2018*). But in the AURORA phase III study, CVC at 12 months
did not improve fibrosis of ≥1 stage in NASH patients with liver fibrosis (*Anstee et al., 2023*).

## Macrophages and skin fibrosis

Keloid (KD) is a common fibroproliferative disease with unknown etiology. It is characterized by the excessive proliferation of fibroblasts and collagen fiber deposition in the healing process of skin injury, which is often accompanied by itching and pain. It is difficult to treat and has a high recurrence rate, which brings a heavy psychological burden to patients.

Skin macrophages include Langerhans cells in the epidermis and BMDM in the dermis. Fibrosis is mainly found in the dermis, and M2 macrophages play a key role in skin fibrosis. Knipper and colleagues have shown that collagen fibril assembly following mammalian dermis injury and repair is highly dependent on M2 macrophages (*Knipper et al., 2015*). Compared with normal skin and scar tissue, M2 macrophages in keloid significantly increased (*Direder et al., 2022*; *Feng et al., 2022*), and promoted the proliferation and migration of skin fibroblasts by generating connective tissue growth factor and activating ERK1/2/STAT3 and AKT/STAT3 signaling pathways (*Zhang et al., 2021*). tsRNA-14783 participates in KD formation *via* promoting M2 macrophages polarization (*Wang & Hu, 2022*). IL$_{13}$RA2 downregulation, a 'decoy' receptor of IL13, in fibroblasts, promotes M2 macrophage polarization and KD fibrosis *via* STAT6 activation (*Chao et al., 2023*). Macrophages and skin fibroblasts were mutually activated to secrete IL-6 and TGF-$\beta$ and promote the fibrosis process through STAT3 phosphorylation (*Bhandari et al., 2023*). HPH-15, a histidine pyridine-histidine ligand derivative, alleviates bleomycin-induced mouse skin fibrosis by inhibiting the phosphorylation of Smad3 in skin fibroblasts and macrophages (Fig. 1) (*Luong et al., 2018*). In addition, CD301b+ macrophages produce PDGF and insulin-like growth factors, which selectively promote AP proliferation and adipocyte-myofibroblast transformation at the trauma site, and then secrete ECM and collagen fibers to promote the fibrosis process (Fig. 5) (*Shook et al., 2018*). In skin fibrosis, macrophages tend to polarize into M2 type, which secrets pro-fibrotic factors to change in the local microenvironment and further promote M2 macrophage polarization.

## Macrophages and cardiac fibrosis

Cardiac fibrosis is the differentiation and proliferation of cardiac fibroblasts and excessive deposition of ECM, leading to cardiac hypertrophy and reduced diastolic function, eventually leading to heart failure. It is a key prognostic factor of heart disease. In the early stage of injury, M1 macrophages are used to induce inflammation and transition from pro-inflammatory M1 to reparative M2 to mitigate cardiac dysfunction after myocardial infarction (MI). In the later stage of injury, M2 macrophages primarily induce cardiac fibrosis. In the fibrotic area of MI, BMDMs differentiate into a-SMA + fibroblasts and coronary artery endothelial cells undergo an EndoMT induced by TGF-$\beta$, which further increases the number of fibroblasts (*Zeisberg et al., 2007*). M2 macrophages aggregate and activate to promote cardiac fibrosis in an angiotensin II-induced hypertensive cardiac model (*Yang et al., 2012*). Similarly, in elderly mice, aldosterone-exposed mice as well as

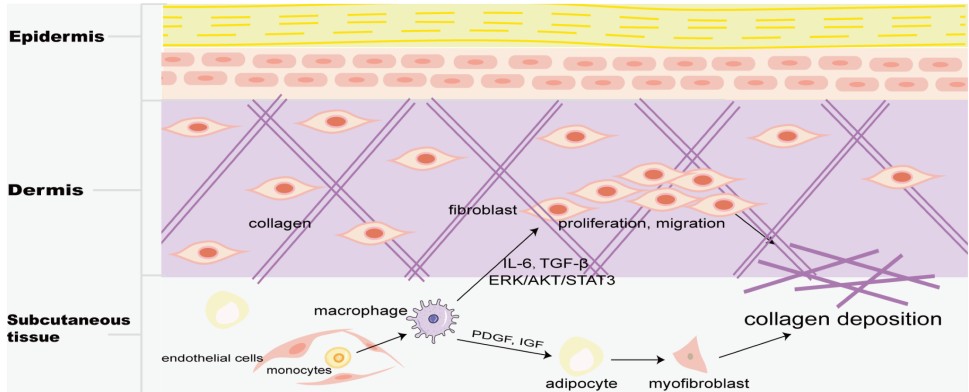

**Figure 5  Macrophages in skin fibrosis.** M2 macrophages not only secrete cytokines such as IL-6 and TGF-β to promote the proliferation and migration of skin fibroblasts, but also produce PDGF and IGF to transform adipose cells into myofibroblasts, resulting in excessive deposition of collagen, and eventually leading to skin fibrosis.

cardiac biopsy specimens of patients with left ventricular ejection fraction retained, M2 macrophages increased and secreted IL-10 to activate fibroblasts and promote collagen deposition and myocardial fibrosis (Fig. 6) (*Hulsmans et al., 2018*). In addition, *Shiraishi, Yamaguchi & Suzuki (2022)* demonstrated that neuregulin 1 (Nrg1) produced by BMDM and Nrg1 co-receptor ErbB expression on the surface of cardiac fibroblasts increased after MI, which combined to activate the downstream PI3K/Akt pathway, inhibit the aging and apoptosis of cardiac fibroblasts, promote their proliferation and lead to fibrosis. On the contrary, some studies have found that the transformation of macrophages from the pro-inflammatory M1 to the anti-inflammatory M2 can alleviate cardiac fibrosis, and M2b macrophages can improve cardiac fibrosis in rat models of myocardial ischemia/reperfusion injury (*Li et al., 2021*; *Wang et al., 2022a*).

## DISCUSSION

The response of macrophages to polarization in different states is the key to their high heterogeneity and functional diversity. The exact mechanism of macrophage polarization in different tissue fibrosis has not been fully elucidated. This article reviews the role of macrophage polarization in different tissue fibrosis and several specific pro-fibrotic macrophage subtypes in recent years. M1 macrophages and M2 macrophages are the two most studied subtypes. It found that the number of M2 macrophages is positively correlated with the severity of fibrosis. M2 macrophages can also transform into myofibroblasts or promote myofibroblasts proliferation to accelerate fibrosis. Blocking the signaling pathways that drive M2 macrophage polarization or targeting directly M2 macrophages are likely to relieve the progression of fibrosis. But there are some unclear insights. On the one hand, it is not absolute whether any form of macrophage is beneficial or disadvantageous for tissue fibrosis. Such as M2 can suppress cardiac fibrosis remodeling after MI (*Li et al., 2021*). On the other hand, MMT is controversial because it is difficult to ensure myofibroblasts

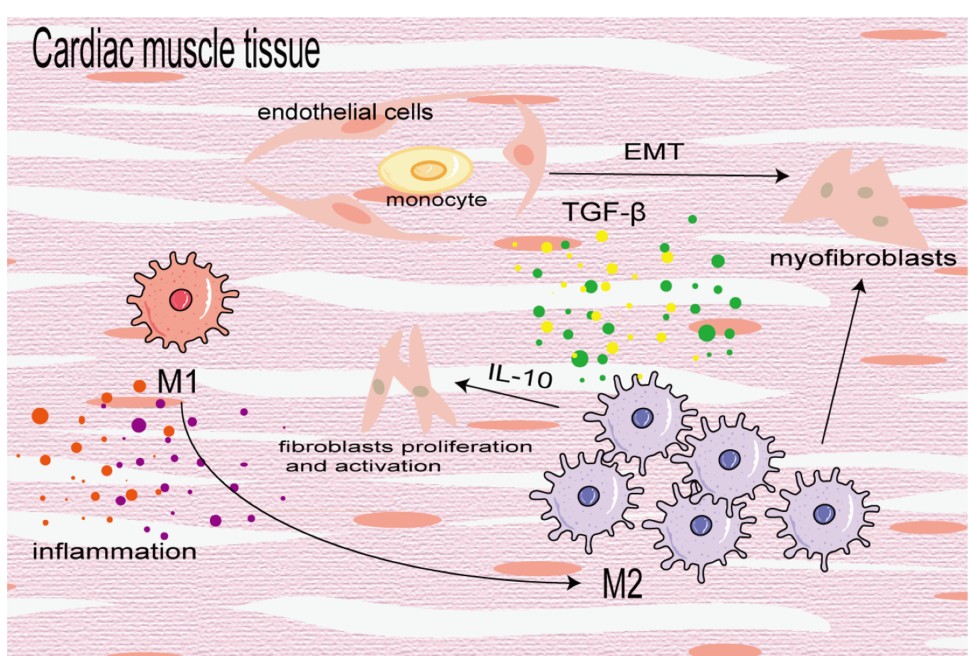

**Figure 6 Macrophages in cardiac fibrosis.** M1 macrophages produce inflammatory cytokines to promote inflammatory response, which can also be converted into M2 macrophages to participate in the process of fibrosis. M2 macrophages differentiate into myofibroblasts, activate cardiac fibroblasts and promote coronary endothelial cells to transform into myofibroblasts.

come from MMT and quantify MMT as a source of myofibroblasts. It is also unclear whether MMT cells make a significant contribution to collagen deposition in fibrosis (*Kramann et al., 2018*). Thus, accurate identification of pro-fibrosis macrophages is very important. Satoh and colleagues found a new class of pro-fibrotic macrophages, which is $Ceacam1^+Msr1^+Ly6C^-F4/80^-Mac1^+$ monocytes and named segregated-nucleus-containing atypical monocytes(SatM) (*Satoh et al., 2017*). It can be considered as a specific subgroup in M2. In the future, we need to identify "disorder-specific monocyte/macrophage subtypes" corresponding to certain diseases and develop novel, more specific therapeutic targets with fewer side effects.

## SURVEY METHODOLOGY

To summarize the role of macrophage polarization in tissue fibrosis from multiple perspectives, we incorporated Medical Subject Headings (MeSH terms) into the PubMed search strategy to search the literature. Search terms included ("Macrophage activation" [MeSH terms] OR ("macrophage" AND "polarization") OR "M1 macrophage" OR "M2 macrophage" OR "myofibroblast") AND "fibrosis" [MeSH terms]. In the process of summarizing the literature on tissue fibrosis, we further refined the tissue classification. We searched the literature with two keywords for fibrosis and macrophages, adding the tissue type ("lung", "kidney", "liver", "skin", or "cardiac",).

## ACKNOWLEDGEMENTS

The authors thank their laboratory colleagues for their help in this review.

### Funding

This study was supported by the Key Scientific Research Project of Medical Science of Shanxi Province (2021XM08), Basic Research Youth Project of Shanxi Province (202103021223442), and the 2020 Shanxi Province Emerging Industry Leadership Project (2020-15). The funders had no role in study design, data collection and analysis, decision to publish, or preparation of the manuscript.

### Grant Disclosures

The following grant information was disclosed by the authors:
Key Scientific Research Project of Medical Science of Shanxi Province: 2021XM08.
Basic Research Youth Project of Shanxi Province: 202103021223442.
2020 Shanxi Province Emerging Industry Leadership Project: 2020-15.

### Competing Interests

The authors declare there are no competing interests.

### Author Contributions

- Huidan Yang conceived and designed the experiments, performed the experiments, analyzed the data, prepared figures and/or tables, authored or reviewed drafts of the article, and approved the final draft.
- Hao Cheng conceived and designed the experiments, performed the experiments, analyzed the data, authored or reviewed drafts of the article, and approved the final draft.
- Rongrong Dai performed the experiments, analyzed the data, prepared figures and/or tables, and approved the final draft.
- Lili Shang performed the experiments, analyzed the data, authored or reviewed drafts of the article, and approved the final draft.
- Xiaoying Zhang conceived and designed the experiments, authored or reviewed drafts of the article, and approved the final draft.
- Hongyan Wen conceived and designed the experiments, authored or reviewed drafts of the article, and approved the final draft.

### Data Availability

This article is a literature review.

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
