# Peer review of "Macrophage polarization in tissue fibrosis"

_PeerJ, doi:10.7717/peerj.16092_

## Round 0.1 · original submission · Major Revisions

The author must modify the manuscript according to the reviewers' requirements.

Reviewer 1 ·

Basic reporting

The authors reviewed the role of macrophage polarization in tissue fibrosis. However, macrophage and different fibrotic diseases are widely reviewed (such as PMID: 24412603, 34093584, 33670759). Besides, this manuscript was not well written, some sentences are confusing to readers. Moreover, the format of some Latin letters and the references are not correctly used. Here are some suggestions of mine:
1. Since this manuscript are reviewing macrophage polarization in tissue fibrosis, the content of macrophage polarization needs to be expanded.
2. The English language and grammar should be improved (such as: line 99-101 sentence unfinished).
3. Format of Latin letters should be check again (such as: line 184 ‘β’, line 190 ‘α’ etc.)
4. Line 317-330 This manuscript didn’t conduct any experiments, how the ‘experiments designed, performed and data analyzed’?
5. References format should be checked carefully. Endnote are recommended to insert citation.

In conclusion, this manuscript is not qualified for publication now.

Experimental design

No comment.

Validity of the findings

No comment

Cite this review as

·

Basic reporting

The review is within the scope of the journal, however, it could benefit from inclusion of more information from primary sources of fibrotic diseases.

This field has been reviewed recently in - 'Monocyte and macrophage derived myofibroblasts: Is it fate? A review of the current evidence' https://doi.org/10.1111/wrr.12946
The present review includes information regarding therapeutic targets of fibrosis and mentions some putative therapeutics under investion. However, it can be significantly improved by including more information. Below are my comments regarding the Introduction -

Major Comment

1. It would be useful to include in the Introduction, towards the end of the final paragraph, what the review is going to discuss (signaling pathways relevant to macrophage driven fibrosis and tissue types most affected by fibrotic disease, potential therapeutics targets and investigational drugs currently in development, etc). The goals of the review, should be outlined clearly.

2. I recommend the authors explain briefly what MMT is in a few sentences, in the introduction.
There is insufficient evidence to confirm how macrophages/monocytes to myofibroblasts transition or MMT occurs. MMT is still a theory. Authors should discuss this in the Introduction briefly.

3. Authors should include statistics on incidence, morbidity and mortality rates of fibrosis and different fibrotic diseases - it may be useful to include a brief paragraph on this, with quantification of the cost of the 'heavy burden on humans' (line 50).


Minor Comments -

1. Sentences in Abstract and Introduction on lines 14 and 27 are too similar. Please consider revising the beginning of the abstract.

2. In lines 29-30, I am not sure how accurate these are, as monocyte differentiation can lead to various types of macrophages.

3. After lines 57, I recommend including a sentence or two regarding the consequences of mild fibrosis?

Experimental design

Major Comments -

1. Throughout the manuscript, there are several places where it is difficult to understand what the authors are trying to convey. I highly recommend the authors to employ a writing service and seek assistance of fluent English speaker to edit the manuscript to improve sentence structure, clarity and grammar.
For e.g., the section in Introduction from lines 33-40 can be improved by breaking down into 3-4 sentences and correcting for grammar.
Also, in lines 61-65, it is not clear what the authors are trying to convey. This section could be broken into more than one sentence and needs more clarity and citations.
The sentences in lines 88-93 could be restructured to improve clarity and should also be cited appropriately.


2. Throughout the manuscript, I felt there are several places that are not adequately cited. I have marked them using annotations.
Also, more than once authors mention 'several studies' or 'many studies' but only include 1 citation (for e.g., lines 121-123). I recommend authors include the citations of all the primary sources that they are referring to.

Minor Comment

Authors could include more terms in their pubmed search with correctly spelled "macrophage polarization', 'M1 macrophage', 'myofibroblast', 'fibrotic diseases', 'tissue fibrosis'

Validity of the findings

Below are my comments on the individual sections and conclusions -

Section 2
Major Comments
1. In section 2.1, Can the author comment on Smad2 as a drug target?
It is important to include and discuss data from in-vivo studies that have successfully decreased or halted tissue fibrosis through targeting of TGF-beta1.

2. On line 99, is Ly2109761 a small molecule drug, or a peptidomimetic or biologic?
What stage of clinical development is this molecule?
Authors should very briefly include these details.

3. In section 2.3, lines 128-130, Authors should specify if the mentioned inhibitors are specific or preferential to a particular JAK or STAT protein, such as JAK2, and include citations next to each inhibitor mentioned.
Reference 19 mentions Cucurbitacin-B, which is known to inhibit Jak2 phosphorylation. Please check if calinurin-B is a typo.
Authors can also include this reference regarding how Cucurbitacin modulates M2 polarization through inhibition of Jak2/Stat 3 signaling. https://doi.org/10.1016/j.jep.2021.114915

Minor comments
1. I recommend authors discuss briefly – the Notch signaling pathway being involved in myofibroblast generation in section 2.
2. In lines 94-95, authors state – “TGF-b3/Smad signaling pathway is one of the main pathways involved in fibrosis.” I think this sentence could be moved to the beginning of the section.
3. In line 96, is the first mention of MMT and authors use abbreviation. Generally, the first mention should be elaborated and from then on, it can be abbreviated. Similarly, please expand EMT in line 99.
4. Regarding line 109- authors should change “(seven-time transmembrane protein)”. Frizzled is ‘an atypical G protein with 7 trans-membrane domains’. Please bear in mind that most all G proteins have 7 transmembrane domains.
5. On line 130, Authors should specify which animal model.
6. On line 143, please name specific inflammatory factors that polarize macrophages into M1 type macrophages.


Section 3
Major Comments
1. In section 3.1, Please include Tacrolimus in the figure with the original paper citation of the study using the drug to target pulmonary fibrosis. Similarly, for section 3.2, please include clodronate in the figure mechanisms. And for section 3.4, include the proposed therapeutic (HPH-15) in the corresponding figure.

2. I recommend the authors include a mention of pericyte trans-differentiation, briefly in a sentence or two in the introductory part, especially that CTGF secretion, can trigger pericytes to transdifferentiate PMID: 23302695.

3. Given that podocytes are similar to pericytes, is it possibly that podocytes may trans-differentiate into myofibroblasts specific to the kidney? There is some evidence to suggest that diseased podocytes aquire pericyte like characteristics and may undergo trans-differentiation into myofibroblasts. PMID: 18337603; PMID: 21931791; PMID: 23325411
In any case podocytes and pericytes can secrete tgf-b1.
4. Treatment with lisinopril as well as Ramipril therapy has also shown to decrease TGFb1 and delay renal fibrosis in mouse models PMID: 12631109. Can authors comment about that, especially because Angiotensin II stimulation also mediates macrophage recruitment and accumulation.

Minor Comment
1. In lines 202 – 208, are these mechanisms of fibrosis specific to renal fibrosis? Please be specific so reader has clarity.
2. In lines 229-231 - this sentence needs citation. Also, it is not clear how chlorphosphonate liposomes alleviate liver fibrosis? Please briefly explain the mechanism (target molecule/signaling pathway).


Discussion
Major comments
1. To me, discussion comes across as a little vague. Perhaps it can be improved by providing specific examples, especially of the therapeutics in various stages of development targeting signaling pathways involved in macrophage polarization that hold promise in treating tissue fibrosis. The authors could also tabulate it.

2. I also recommend briefly discussing, instances or examples where macrophages are not the primary mediators of fibrosis and some of the contention regarding MMT.

Minor Comments
1. Sentence 293 should include specifically why macrophages in tissue fibrosis have attracted attention?
2. Sentences in lines 293-298 could be restructured to remove redundancies and improve readability.
3. On lines 304, authors state – “In addition to clarifying mechanism of different macrophage
subpopulations promote, inhibit, or reverse fibrosis, future research should also elucidate the
timing and target of regulating macrophage polarization and phenotypic conversion.”
I find this sentence confusing. Perhaps it can be broken into two sentences with more specifics such as naming the specific macrophage subpopulations.
To me, it is not evident what the authors refer to by timing.

Cite this review as

---

## Round 0.2 · Minor Revisions

Because the number of reviewers who agreed to re-review was insufficient, I invited more reviewers and two of them have provided reviews. The author should try to modify the manuscript so that the majority of reviewers agree to accept it.

·

Basic reporting

No comment

Experimental design

Authors have satisfactorily responded to my comments

Validity of the findings

Authors have satisfactorily responded to my comments

Additional comments

Authors have done a significant amount of work to substantially improve the manuscript from the first version. In particular, authors are to be commended for including Table 1.
However the article still needs several improvements to improve coherency as well as grammar fixes and corrections to sentence structures.

1. In Section 2.2, authors write “The expression products of these target genes can induce EMT to promote cardiac fibrosis (Tao et al. 2016).”
Authors could describe what the target genes are or what the expression products are.
Similarly, in Section 2.4, authors should include details on the downstream effectors of Notch signaling.

The manuscript should undergo thorough proof-reading and copy editing. I have mentioned some inconsistencies/recommendations below -

1. Abstracts
I recommend “…, and hopefully, provide a useful reference for the future treatment of fibrosis.”

2. Introduction
This sentence on page 1-2 is not clear - “Macrophages can be recruited from Bone marrow-derived macrophages (BMDM) and tissue-resident macrophages (TRM) (Davies et al. 2013; Ginhoux & Jung 2014; Locati et al. 2020).”

3. Do authors mean ’undergo’ instead of ‘of’ “Myofibroblasts of excessive proliferation and activation produce ECM and collagen, which play an important role in the process of fibrosis”

4. This sentence on page 3 appears incomplete towards the end- “Subsequent studies have shown that MMT is involved in progressive fibrotic disease and MMT macrophages mainly from M2 macrophages (Meng et al. 2016; Yang et al. 2021).”

5. Section 2 
On Page 4-5 it is not clear what fisetin is (small molecule, biologic?) and it is implied, but not made explicitly clear that fisetin targets Smad2/3.
“Smad2/3 may be a potential target, such as fisetin protected against renal fibrosis by inhibiting the phosphorylation of Smad3, and accumulation of profibrotic M2 macrophages (Ju et al. 2023).”

6. Discussion
In this sentence, “However, MMT is controversial because it is difficult to ensure myofibroblast comes from MMT and quantify MMT as a source of myofibroblast by co-expressing both lineages.”
Do authors mean that quantifying MMT as a source is difficult, or that MMT as a source of myofibroblast should be quantified by co-expressing both lineages? Please clarify.

Cite this review as

Reviewer 4 ·

Basic reporting

The overall language of the manuscript is acceptable, although some sections (commented below) are hard to understand and could be optimized. This review fits within the scope of the journal, and PeerJ has previously published similar reviews (PMID: 36196399). Braga et al. authored a review in 2015 covering similar topics (PMID: 26635814) - it is recommended to consider focusing more on research and perspectives that were not addressed in the previous review and emphasize recent advancements and newly-emerging topics in the field.

Below are some specific comments on basic reporting:
- Line 29-31: "Macrophages can be recruited from Bone marrow-derived macrophages (BMDM) and tissue-resident macrophages". The term "recruited" used in this context may be misleading. Consider using an alternative term like "derived from" instead.
- Line 68: "Epithelial-to-mesenchymal transition (EMT) showed the process to lose epithelial features and acquire mesenchymal features in the treatment with profibrotic factors such as TGF-β (Zeisberg & Kalluri 2008)". This sentence is a bit confusing and could benefit from clarification. The concept of EMT should be introduced first before introducing its relevance to fibrosis. Also, the cited study by Zeisberg & Kalluri appears to be in the specific context of chronic kidney disease and kidney models. It might help to point the readers to additional relevant references. Furthermore, the term "showed" seems inappropriate – something like "describes" or "involves" may work better in this context.
- Lines 128-134 on the description of the Wnt/β-Catenin pathway are missing references.
- The discussion section was a bit hard to follow. I suggest revising the discussion section to enhance clarity and readability. Particularly in lines 351-354, where it was first talking about M1/M2 classification, then immediately jumps into MTT, then scRNA-seq. There was an absence of a transition. There is also insufficient reasoning or evidence presented in the discussion.
- Please review the text labels in the figures presented to make sure that they are clear. Some text labels in Figure 2-3 were small – larger labels help provide clarity.
- Please check the label names in Figure 2 and ensure that they are in English. Also, note the extra label "Fibroblast" outside the image.
- Lines 255-257: "Cirrhosis affects approximately 2.2 million adults in the US. From 2010 to 2021, the annual age-adjusted mortality of cirrhosis increased from 14.9 per 100 000 to 21.9 per 100 000 people (Tapper & Parikh 2023)" appears to be copied word for word from Tapper & Parikh 2023 without marked as a quote. Please paraphrase or mark it as a quote.

Experimental design

- The current survey methodology, which relies solely on keyword searches, may not capture all relevant literature. To potentially enhance the comprehensiveness and accuracy of the search, it may be beneficial to incorporate Medical Subject Headings (MeSH terms) into the PubMed search strategy. For instance, using terms like 'Macrophage Polarization'[Mesh] AND 'Fibrosis'[Mesh] may help retrieve more relevant articles on the topic.
- It may be helpful to clarify the reason for selecting to focus on lung, kidney, liver, skin, or cardiac tissue fibrosis.

Validity of the findings

- When summarizing the findings of cited studies, I encourage the authors to do a more critical evaluation of these findings. This includes discussing the implications and limitations of each finding, noting potential controversies, and identifying areas that require further research. It may also be worthwhile to highlight studies that have agreeing or disagreeing viewpoints on the same topic. This will greatly improve the value of this review and will provide a more comprehensive perspective on the state of the field.
- When reporting findings from the literature, it would be helpful if the authors could provide overall a bit more details about the context and the experimental approaches used in the cited studies. This would enable readers to evaluate the findings and understand their relevance to the reviewed topic. For example, lines 201-204: "In fibrotic lung tissue 202 with unilateral ureteral obstruction (UUO) rats, approximately 30% of myofibroblasts were CD68+³-SMA+ MMT cells, and up to 35% were co-expressing for M2 macrophage marker CD206 (CD206+³-SMA+) (Yang et al. 2021)". It may be worth mentioning that Yang et al. used immunofluorescence staining as their bases for quantification. Including such information may enhance the transparency and interpretability of the literature review.

Cite this review as

Reviewer 5 ·

Basic reporting

Well written review, english is clear and unambigous

Experimental design

No comments

Validity of the findings

Well written Review

Additional comments

Title : Macrophage polarization in tissue fibrosis (manuscript #81507)

Yang et al., had written the review-manuscript well and its interesting to the general audience in the field of immunology to understand the role of macrophages in tissue fibraosis. However, a few comments should be addressed to have it published.
Minor Comments
Abstract
Line #24. Hoping to provide a useful reference…..rewrite to Hoping to provide useful review
Introduction
Line # 33: Activated or M1 macrophages ….rewrite to activated or M1 macrophages that are
Line # 39-40…Rephrase the sentence to make more readable and for clarity
Line #95- 102- Survery Methodology entire paragraph to be moved to end of the discussion above the acknowledgement…the conitunity of the raeding will be lost if placed between introduction
Line #204-206- Rephrase the sentences
Line #259- BMDM?? Fully annotate the name
Linen#333- Nrg1---annotate the protein name
Discussion
Line #342: Recently….functions…. rephrase the sentence for readability.
Major Comments
No Major comments

Cite this review as

---

## Round 0.3 · Minor Revisions

Please modify the manuscript as required by the reviewer.

·

Basic reporting

no comment

Experimental design

no comment

Validity of the findings

no comment

Additional comments

Authors have significantly improved the manuscript since the first round of review. Authors have also satisfactorily responded to all of my comments. I have a few minor suggestions/corrections -

- Line 29-31: 'Macrophages can be bone marrow derived (i.e., Bone marroe derived macrophages or BRDM) or tissue resident (i.e., tissue resident macrophages or TRM)

- Line 54: ';and the severe fibrosis ...' could be replaced with '; while the severe fibrosis'

- The sentence on lines 58-62 "It is highly heterogenous ....."
Assuming that "It" refers to MMT, this sentence should come after the first mention of MMT
If "It" refers to fibrosis, then "such as" should be changed to "as" or "because"

-Line 114: "in vitro experiments" should be replaced by "in in-vitro experiments"

- Line 118: BLM should be explained at its first ocurrence in the text

-Line 171: Please italicize 'Schistosoma japonicum'. Please also mention in parenthesis that it is a parasite.

Line 206: Please remove 'activate', as it may be redundant

-Line 222: Please clarify what circulation cell migration is.

-Line 232: Please remove 'be'

-Line 314: Please abbreviate bleomycin as BLM at first instance and use BLM from then on, or use 'Bleomycin' throughout the manuscript. Either way, please be consistent.

Cite this review as

Reviewer 4 ·

Basic reporting

Comments from the previous round of review were addressed.

Experimental design

no comment

Validity of the findings

no comment

Cite this review as

Reviewer 5 ·

Basic reporting

no comments

Experimental design

no comments

Validity of the findings

no comments

Cite this review as

---

## Round 0.4 · accepted · Accept

After the author's revision, the quality of the manuscript has been greatly improved. It has been endorsed by a sufficient number of reviewers, and I also agree that this paper meets the requirements for publication.